# Diagnostics of Allergy to Furry Animals—Possibilities in 2024

**DOI:** 10.3390/jcm13113239

**Published:** 2024-05-30

**Authors:** Tomasz Rosada, Kinga Lis, Zbigniew Bartuzi, Magdalena Grześk-Kaczyńska, Natalia Ukleja-Sokołowska

**Affiliations:** 1Department and Clinic of Allergology, Clinical Immunology and Internal Diseases, Ludwik Rydygier Collegium Medicum in Bydgoszcz, Nicolaus Copernicus University in Toruń, 87-100 Toruń, Poland; kinga.lis@cm.umk.pl (K.L.); zbigniew.bartuzi@cm.umk.pl (Z.B.); ukleja@10g.pl (N.U.-S.); 2Clinic of Allergology, Clinical Immunology and Internal Diseases, Jan Biziel University Hospital No. 2 in Bydgoszcz, 75 Ujejskiego St., 85-168 Bydgoszcz, Poland; magdalenagrzesk@gmail.com

**Keywords:** allergy, furry animals, diagnostics, guidelines, allergy tests

## Abstract

Diagnosing allergies is not always easy. Sometimes the symptoms reported by the patient do not match the results of diagnostic tests. Diagnostics of IgE-dependent allergy, including allergy to furry animals, can be based on two basic strategies, “BOTTOM-UP” and “TOP-DOWN”, and almost all available allergy tests can be used ranging from skin tests, through specific IgE, to molecular panels and challenge tests. Currently, most information regarding the diagnosis and incidence of allergies to furry animals focuses almost exclusively on the two most common pets—dogs and cats. This is certainly due to the fact that allergy to cats and/or dogs is a common phenomenon, has a huge impact on quality of life, and is a challenge for doctors of many specialties. However, the growing number of owners of other pet species means that diagnostic methods must keep up with the changing needs of patients. Further research in these groups will certainly provide new facts and facilitate clinical decision-making when working with allergic patients.

## 1. Introduction

Allergy diagnostics, including allergy to furry animals, is not always a simple matter. Every now and then, the symptoms reported by the patient do not match the results of diagnostic tests. One of the most fundamental aspects for the patient is predicting the coexistence of allergy to other animals, as well as evaluation of the natural course of the allergy. It is not always easy to clear up these doubts [1].

The choice of diagnostic proceedings depends on several facets—on the one hand, an individual approach to each case is of major importance, evaluating which diagnostic method will allow for early diagnosis and at the same time cause the lowest stress for the patient. On the other hand, the cost and accessibility of a diagnostic method is essential. All of the abovementioned aspects are worth taking into account when selecting a particular diagnostic model [2,3].

In order to prepare this article, the Google Scholar and PubMed databases were searched using the terms “allergy”, “fur animals”, and “diagnostic”. The following article is *a review* and does not meet the criteria of *a systematic review*.

## 2. Basic Diagnostic Models in Allergology

Allergy diagnostics of IgE-dependent can rely on two basic strategies, i.e., BOTTOM-UP and TOP-DOWN (Figure 1). It is obvious that not every clinical case (patient) can simply be classified into one diagnostic model; however, general guidelines of the said strategies could prove useful in everyday medical practice [4].

TOP-DOWN is a diagnostic approach predominantly focusing on the symptoms that made the patient seek help. On the grounds of an in-depth interview, thorough physical examination, and an attempt to demonstrate allergen–symptom correlation, diagnostic tests are selected (either in vivo or in vitro) for single allergens that are marked out for being the most possible causative agents for the disease. Subsequently, and in line with the top-down principle, the pool of suspected allergens is narrowed down until eventually, when possible, trials are conducted so as to single out specific allergenic components responsible for triggering clinical symptoms. It is worth mentioning that the first examination in the said model is most often allergen extract-based skin tests that do not reveal the answer as to which specific molecule/particle triggers the allergy pathomechanism, and what cross-reactivity might be anticipated. Doctor’s experience is of utmost value in this diagnostic model aiming to rule out randomness and to accelerate both the diagnosis and treatment implementation process [4,5,6,7,8].

On the other hand, the BOTTOM-UP strategy can be perceived as a specific reversal of the TOP-DOWN diagnostic rules. This modelling begins with attempting to demonstrate the specific allergen profile of a patient as broadly as possible, including specific allergenic components, and only then is the medical history of a given patient analysed based on available results of antigen-specific IgE (asIgE) concentrations to specify further actions. Eventually at this stage, presence of asIgE is correlated with the onset of clinical symptoms and allergy is distinguished from atopy [4,5,7,8].

In recent years, another diagnostic scheme has been proposed that seems to combine those presented above, i.e., U-shaped molecular diagnosis. It assumes starting the diagnosis by combining clinical data with molecular tests (descending arm), and then continuing the diagnostic process from molecular data to specific clinical implications (ascending arm), taking into account potential cross-reactions. In this scheme, molecular diagnostics may be based on both the determination of selected, single asIgE (singleplex) and the use of large molecular panels, but the selection of the method should always result from the clinical situation and be adapted to the expectations set for molecular diagnostics in a given case (high sensitivity vs. number of allergens tested) [4].

## 3. Review of Commercially Available Allergy Tests Useful in the Diagnostics of Allergy to Furry Animals

Table 1 presents the main diagnostics allergy tests, including their division into in vitro and in vivo.

Skin tests are the most commonly performed tests in allergy diagnostics. They are the cornerstone of diagnostic evaluation in the TOP-DOWN model and allow for preliminary verification of findings from medical history and physical examination. We distinguish several types of skin tests: skin prick tests, prick-by-prick tests, intradermal tests, and epidermal patch tests.

The skin prick test (SPT) is one of the oldest methods of allergy diagnostics. It is used in diagnosing IgE-mediated allergy based on the mechanism of Type I hypersensitivity by Gell and Coombs. Allergen extracts, used for skin prick tests, are a specific mixture of various molecules, a number of which may have allergenic potential.

The standard of performing and interpreting the results of skin prick tests must be consistent with the current clinical practice guidelines described more extensively in separate papers [11,12].

Based on the Register of Medicinal Products, as of 31 December 2023, in order to perform skin tests, the Polish market currently offers the allergen extracts manufactured by three companies: Allergopharma GmbH & Co. KG (Reinbek, Germany), Stallergenes S.A. (Baar, Switzerland), and HAL Allergy B.V. (Leiden, The Netherlands) [13]. Table 2 below shows all available allergen extracts in diagnosing allergy to furry animals.

In the standard set of airborne allergens according to the recommendations by the European Academy of Allergy and Clinical Immunology (EAACI) [11] and the American Academy of Allergy, Asthma, and Immunology (AAAAI) [17], there are only dog and cat fur extracts, therefore the most common pets. Other extracts are commercially available, but relatively seldom used in daily medical practice.

Skin prick tests are the basis of allergy diagnosis, can be performed in an ambulatory setting with the result in 15 min. This inexpensive diagnostic method has many disadvantages and there is often a need of further diagnostics. The SPT can only confirm the presence of asIgE for a given allergen source in the patient’s plasma, but it cannot specify which particular molecule, i.e., allergenic component, is responsible for inducing the whole pathomechanism. This, clearly, has clinical implications, such as the inability to extrapolate possible cross-reactions [11,18,19,20]. Only original, standardised extracts can be used for testing, which are stored in line as recommended by the manufacturer (usually at a temperature of approx. 4 °C) and are subject to periodic checks in order to verify their further clinical suitability [14]. Curin et al. described the differences in extracts of commercial skin prick tests. The comparison revealed a 20-fold variation regarding the total protein content and lack of Can f1 and Can f2 in one of the extracts. The contents of the major dog allergen Can f 1 and of Can f 2 varied between the extracts [21]. This resulted in the inaccuracy of the SPT and ruin the diagnostics process of allergy.

It is also highly important to prepare the patient properly for these tests. In order to recognise the results of the examination as reliable, the patient must follow the pharmacological regimen, i.e., stop taking drugs that might affect the above-defined biological processes. In the case of clinical suspicion of allergy to specific furry animals, primarily on the grounds of medical history, skin tests, covering most allergenic components, appear to be a very good choice as a screening test. Nevertheless, it should be noted they will not provide an in-depth analysis, such as predictability of possible cross-reactions [11]. In theory, it is feasible to perform skin tests with particular allergenic components. In research of Gamboa et al., into peach allergen (Pru p 3—lipid transfer proteins), a high sensitivity and specificity of such technique was indicated when weighed against the asIgE concentration and basophil activation test [22,23]. Yet, these tests, at least for allergy to furry animals, are not available in daily medical practice, and due to the likely high cost of preparing recombinant protein for testing, their availability would probably be significantly limited.

Prick-by-prick tests are a very simple diagnostic method resembling the above-defined skin prick test. The only difference is the use of native allergens obtained by injecting the source of the allergen with the same blade (e.g., food, immersion in milk), and next injecting the dermis of the tested patient. Contraindications, restrictions, and interpretation of the results are similar to those of the SPT. In the case of allergy diagnostics to furry animals, extraction of a native allergen can be quite inhibited, especially when testing animal fur [11,18]. Prick-by-prick tests can be prepared by rubbing animal fur with a blade, and then piercing the skin or, after piercing the skin, rubbing the appropriate place on the skin with fur. Prick-by-prick tests, with aeroallergens and food allergens, are characterised by good sensitivity and specificity, and the negative predictive value. However, they do have limitations, the most important of which is the possibility of contaminating animal fur with allergens from other sources [12,24,25,26].

Intradermal tests are usually used to diagnose hypersensitivity to drugs and Hymenoptera venom. They found no wider application in the diagnosis of allergies to furry animals [12].

Epicutaneous patch tests (EPTs)/atopy patch tests (APTs) are fundamental to diagnosing antibody-independent allergy based on the pathomechanism of Type IV hypersensitivity by Gell and Coombs. After a lipophilic allergen of a low molecular weight penetrates deep into the dermis and binds with endogenous protein, a complete antigen is formed (hapten–protein complex). During the test, the dermis is studied for the presence of T lymphocytes sensitised to a given antigen. Formed as a result of the primary immune response, these cells are responsible for inducing a local inflammatory response by secreting a variety of cytokines after the first exposure to the antigen. In addition, due to their cytotoxic properties, sensitised T lymphocytes eliminate the particular antigen. Skin lesions, in the form of erythema, oedema, papules, or blisters, observed as a consequence of the abovementioned mechanisms, usually occur within 24–48 h after having been exposed to the causative agent and indicate a positive test result [27].

Patch tests are usually performed on the back skin; however, in the absence of place or in case of contraindications for testing the site, the arm or thigh skin is used. In the first stage, 20 mg samples of the test substances in Vaseline or single droplets of the substances in liquid are inserted into square chambers, mounted in a set of 10 on a hypoallergenic patch, and then placed on the skin. The square shape of the chambers simplifies the difference between the true allergic reaction (fuzzy angles, rounding edges) and the reaction to irritation (precise reflection of the chamber shape) [28].

In his study, Nečas [29] confirmed that the APT is a helpful tool for identifying airborne allergens as triggering factors of atopic dermatitis (AD). He investigated a cohort of 125 patients with AD who underwent patch testing for selected inhalant allergens. Positive test results for cat dander were obtained in 6 patients (4.8%), for dog dander in 10 patients (8%), and for horse dander in 4 patients (3.2%). These patients experienced AD flares after direct exposure to allergens classified as airborne in the abovementioned animals [25]. Fuiano et al. stress the fact that simultaneous positive results of an SPT, asIgE, and APT indicate the co-occurrence of an allergic reaction of the immediate and delayed type [30].

Allergen-Specific IgE (asIgE) is the most commonly applied method of in vitro allergy diagnostics. The basis of the tests is the biding of asIgE to the patient’s plasma by a specific allergen used in the test. Depending on the implemented method, the allergen can be either suspended in the liquid phase or placed on a coated disc (solid phase). Then, in the case of the presence of asIgE in the plasma of the patient, an allergen–asIgE complex develops, whose manifestation is ascertained using animal detection antibodies directed against solid fragments of human IgE. The results are qualitative or semi-qualitative [31,32].

Allergen extracts or allergenic components (native or recombinant) are used to bind asIgE. Allergen extracts are a mixture of various molecules (including non-allergic) whose composition may vary depending on the native material, the preparation method, or the storage method. Tests based on extracts allow for determining the source of the allergen, but prevent more in-depth analysis, such as an attempt to predict possible cross-reactions, or to estimate the exacerbation reactions after contact of the body with the given allergen. This can be a challenge for the clinician, especially when trying to issue appropriate recommendations for the patient or qualification for treatment, e.g., as part of specific immunotherapy. These limitations can be overcome by means of molecular diagnostics by using allergenic components when determining asIgE, i.e., specific chemical molecules (most often protein, less often carbohydrates) that are known to have allergic potential. Components are elicited by isolating them directly from allergen extracts (native components) or through the use of genetic engineering methods (recombinant components). Molecular diagnostics can be carried out in a limited range by determining single and selected asIgE (singleplex method) or by determining asIgE in a multi-component way for at least two allergens simultaneously in a single test. Thanks to the use of allergen matrices and advances in nanotechnology, multi-component allergy tests allow to elicit an individual, molecular allergy profile of the patient during one study. It should be noted that multi-component and multi-parameter assays are not synonymous, as some commercially available multi-parameter tests use both allergenic components and allergen extracts, and some of them allow for determining the total IgE concentration [4,33].

Table 3 presents the allergens of selected furry animals that have been listed as allergens by the International Union of Immunological Societies (IUIS) taking into account the possibility of marking asIgE both in assays by the ImmunoCAP singleplex method, allergen small panels (POLYCHECK), and in currently available multi-component and/or multi-parameter assays (ImmunoCAP ISAC, ALEX^2^).

It is worth noting that most “animal” allergens are included in the ALEX^2^ multiplex assay, which at the same time is the only test simultaneously evaluating asIgE both against selected allergen extracts and allergenic components (both recombinant and native). In terms of the scope of asIgE diagnostics for furry animals, the second place belongs to ImmunoCAP singleplex (32 possibilities); however, it requires the physician in charge to select diagnostics of specific tests, and it does not allow for carrying out broad molecular diagnostics in an economical way. Yet, ImmunoCAP singleplex is the method with the highest sensitivity and specificity, the gold standard. This reference method is used to compare the results obtained from other certified methods for in vitro diagnostics. Other *singleplex* methods are also available on the diagnostics market, but they are used much less frequently. A thoroughly taken anamnesis and physical examination often allow for identifying the most suspicious extracts and allergenic components. In cases of strong clinical suspicion of hypersensitivity to a particular animal, multiplex assays appear to be unnecessary for establishing the correct diagnosis; though they may provide additional data and sometimes help in dispelling doubts related, for instance, to prevalence of symptoms as a result of allergic cross-reactions. The ImmunoCAP ISAC E112i assay solely contains components (native and recombinant); there are 16 referring to furry animals, and most of them can also be assayed individually as part of the ImmunoCAP singleplex testing. In the small allergen panels POLYCHECK and EUROLINE, allergen extracts prevail, although allergenic components are available for selected allergens. The available research lacks complete and credible reports of allergen-specific IgE results for a number of species of furry animals, both in the singleplex and multi-component/multi-parameter assays. The vast majority of available findings only concern sensitisation to dogs and cats.

The basophil activation test (BAT) is the next and most modern method for diagnosing IgE-dependent allergy. It is based on the molecular mechanism that occurs in a living organism after exposing basophils to an allergen. Despite the fact that under homeostatic conditions, basophils make up merely 0–1% of the entire leukocyte pool in the peripheral blood, they can be responsible for the occurrence of severe allergic reactions, and at the same time allow for carrying out diagnostics, already on the basis of a small volume of the patient’s blood [48,49].

In sensitised persons, B lymphocytes produce allergen-specific IgE, which then binds with surface receptors of cell membranes, among others, basophils (FcεRI). After having been exposed to an allergen, the phenomenon of cross-linking occurs; afterwards, the cell is activated, and the contents of its granularity are released. Basophil activation can be evaluated by determining the expression of markers, e.g., CD63 and CD203c in flow cytometry [48,50]. Basophils can be stimulated with allergen extracts or allergenic components; yet, it is imperative to define the main objectives of the test every time before examination is performed. For instance, the use of allergenic components will allow for identifying a particular molecule responsible for the symptoms of hypersensitivity, and will facilitate, amongst other things, predictability of cross-reactions, but at the same time will eliminate the possibility to determine allergy to other molecules, including those that have not yet been isolated and separated, and that would be present in allergen extracts [35]. Nowadays, there are no explicit instructions as to interpreting the BAT, and it is recommended that each laboratory form their own cut-off points for positive values. Most often, as a positive result, the stimulation index (the ratio of the number of activated basophils after stimulation to the number of activated basophils in a negative control) of a value at least 2 is assumed, or cut-off points are determined (% of basophil stimulation in the treatment group against the control group) [31].

Today, the basophil activation test is used for diagnosing acute immediate allergic reactions—primarily to drugs, foods, and Hymenoptera venom resulting from activated mediators released by basophils and mast cells—after exposure to an allergen. In view of the fact that blood basophils are significantly more accessible to examination than tissue mast cells, they are the “material of choice” for allergy testing. The BAT is characterised by very high sensitivity and specificity, and given the nature of the study, it is sometimes referred to as an “oral challenge in a test tube”. It is worth highlighting that the BAT is a study of a considerable importance to monitoring of immunomodulatory therapy, as it allows for assessing its efficacy as early as at the cellular level [51,52].

Käck et al. were probably the first to attempt to use the BAT to distinguish symptomatic from asymptomatic people with serum asIgE for dog dander. The study incorporated 58 children aged 10–18 years, of whom 69% reported symptoms of allergic rhinitis, and 52% symptoms of dog-induced asthma. Blood samples were assessed against the presence of asIgE to dog dander extract and to selected molecules Can f 1 to 6, and the presence of IgG and specific IgG4 to dog dander extract, Can f 1, Can f 2, and Can f 5. The basophil activation test was performed with dog dander extract and with two molecules, for which elevated levels of asIgE were elicited in a given patient. All patients reporting rhinitis (*n* = 40) or asthma (*n* = 30) after exposure to dog allergens showed a positive BAT response to dog dander allergens. No patient with a negative BAT response to dog fur (*n* = 4) reported manifestations of respiratory disorders after contact with a dog. Furthermore, the authors stressed that the BAT can be a useful complementary tool to reduce overdiagnosis of a dog allergy, whilst the presence of IgG and IgG4 antibodies may reflect vulnerability to dogs, rather than tolerance to their allergens [53].

The available literature lacks research into the efficacy and cost-effectiveness of allergy diagnostics to furry animals using the basophil activation test. As of this moment, when it comes to furry animals, the BAT is run first and foremost in scientific research and most commonly concerns canines and felines. It appears that with the enhancement in diagnostic standards using the BAT, increased availability for testing and reduced costs, the said test, as a test tube challenge, is going to be an indispensable alternative to standard provocation tests, also in diagnosing allergies to furry animals.

The extracts and allergenic components, available for the BAT assay, which may be relevant to allergies to furry animals are presented in Table 4 [54].

The mast cell activation test (MAT) is another attempt to refine the diagnostics of IgE-mediated allergy and reduce false positives. Bahri et al. proposed the mast cell activation test. In their experiment, they used human mast cells from the peripheral blood of non-sensitised donors, which, after preliminary culture under standardised conditions, were sensitised with serum from patients allergic to peanuts. The sensitised mast cells were then stimulated with standardised solutions of peanut allergens. After allergen stimulation, the intensity of activation in the sensitised mast cells was assessed [55].

In accordance with the MAT guidelines, human mast cells, taken from healthy donors, are cultured in standard laboratory conditions, and afterwards sensitised with serum from the tested patient. At the sensitisation stage, asIgE expressed in tested serum binds to FcεRI on the surface of cultured mast cells. Sensitised mast cells are then stimulated by the tested allergen. If the test plasma contains IgE specific to the tested allergen, mast cells are activated by an allergen and degranulation occurs. The degranulation process can be evaluated with the use of immunochemical techniques by measuring the concentrations of prostaglandin D2, β-hexosaminidase, and the intensity of expression of specific classification determinant (CD63, CD107a). With reference to the BAT, the possibility of using fresh frozen plasma is essentially a benefit of the MAT, which can be safely transported to reference centres [55,56].

The literature concerned with the matter lacks data on attempts to use the MAT in diagnosing allergy to furry animals.

Provocation Tests: In terms of allergology, provocation tests comprise the intravital testing of the response exhibited by individual organs after exposure to culprit allergens provoking a specific clinical manifestation. Depending on the nature of allergen (foods, aeroallergens, drug, etc.) and characteristics of reported symptoms, various diagnostic routes of exposure are selected, i.e., oral, nasal, conjunctival, and bronchial testing. Though provocation tests are considered a gold standard for diagnosing allergy, they are relatively less commonly performed because they entail high risk of adverse effects, considerable cost, and time consumption. The critical factor excluding the patient from qualifying for provocation tests is resolved anaphylaxis after exposure to the culprit allergen, as in that case, a risk for the patient disproportionately exceeds the expected diagnostic yield of the test. What also needs to be considered is drugs taken by the patient that can impact the test result, chronic diseases, or acute infectious conditions developed in the patient. Principal reasons for recommending provocation testing are in the case of ambiguous results from other allergy tests, and as an attempt to determine whether the treatment applied succeeded in eliciting tolerance to a given allergen [9,57,58].

Currently, there are no unequivocal standards for carrying out and interpreting the results of provocation tests. Most research relates to provocation with food and drugs, while it is difficult to find publications concerning provocation tests using furry animal allergens. In the case of allergies to furry animals, it appears that nasal and bronchial provocation tests could be of greatest relevance since the vast majority of animal allergens are airborne allergens, and manifestations associated with them most commonly affect the respiratory tract.

Given the proliferation of most animal allergens, a provocation test using an allergen challenge chamber (ACC) may turn out an ingenious solution. This method of provocation seems to be a gold standard for aeroallergen provocation. The ACC is a specifically designed device enabling exposure to a controlled environment in which the patients respond to provocation with an airborne allergen of a specific concentration. The concentration should be sufficiently high to induce a mild allergic reaction and needs to be maintained at a steady level for several hours. Employment of a stable and repeatable allergen load ensures balanced symptom assessment. The ACC allows for simultaneous provocation in the upper and lower respiratory tract, eyes, and skin. During examination, it is feasible to elicit subjective assessment of the patient’s health condition, and objective diagnostic test results, e.g., spirometry, rhinomanometry, or laboratory tests. All ACCs are approved for pollen provocation (grass, birch, ragweed, Japanese red cedar), and some are approved for house dust mite and cat allergens [59,60,61].

Hossenabaccus et al., in their publication on the study of cat allergy using a controlled methodology, list three possible ways of provocation with the allergen: placement of the patient in a cat room—with natural exposure, use of allergen exposure chamber, and nasal challenge. Natural exposure cat rooms played an essential role in establishing the foundation for understanding cat allergen-triggered allergic rhinitis. Numerous limitations on the said study, including variable allergen ranges and various study designs, are evidently indicative of the need for a more standardised protocol. Cat rooms are usually small rooms containing 1–2 neutered cats with a litter tray or without one. Directly during the provocation, cats are either kept in cages inside the room or they move freely around the room. Unfortunately, cat rooms are characterised by various allergen exposures depending on the design of the facility, and the allergen range in the same room within the course of the experiment is variable, which may considerably affect the results obtained [62,63,64].

## 4. Cross-Reactions to Furry Animal Allergens

Cross-reactions play a key role in interpreting the results of diagnostic tests. Knowledge of cross-allergy mechanisms facilitates predictability as to which allergens can induce clinical manifestation, even in the case of primary exposure. In this respect, the results of multi-component studies seem to be highly important, especially when formulating recommendations to the patient and selecting the appropriate immunomodulatory therapy.

Cross-reactions result from the affinity of two allergens recognised by the same asIgE. So far, much attention has been devoted to affinity of the primary structure of allergenic proteins. However, currently, both linear and conformational epitopes are being investigated in the context of cross-reactivity. It is argued that two proteins within amino acid composition share approx. 70% identity, and this creates the possibility of an asIgE “error” and the occurrence of cross-reactions [65].

Possible cross-reactions within the group of furry animal allergens have been collected and are shown in Table 5. Lipocalins, as the primary allergen proteins from furry animals, have a distinctive characteristic of high cross-species affinity and seem to be accountable for most cross-reactions in the group described. Several studies have demonstrated that polysensitisation to furry animal-derived lipocalins is associated with more severe allergy symptoms involving the respiratory tract, an increase in concentrations of asthma biomarkers, and acute functional respiratory disorders. The term “molecular spreading” has been introduced to highlight this phenomenon [66]. It is clear that one group of allergens does not exhaust all possibilities of cross-reactivity, all the more so as it has already been mentioned; nowadays, greater significance is attached to the conformational similarity rather than to the identity of amino acid composition [67]. Cross-reactivity of IgE does not always indicate clinical cross-reactivity. In some clinical situations, it is important to identify the primary source of the allergen, particularly if a specific immunotherapy is scheduled.

In their study, Hemmer et al. revealed that double-sensitisation to cat and dog, and cat and horse, and polysensitisation were associated with an increasing prevalence of the cross-reactive lipocalins Fel d 4/Can f 6/Equ c 1 and Fel d 7/Can f 1. Although these lipocalins were not reliable markers for genuine sensitisation, the comparison of sIgE levels may help to diagnose and find a primary sensitiser [68].

**Table 5 jcm-13-03239-t005:** Cross-reactions in allergen groups of furry animals [4,69,70,71,72,73,74,75,76,77,78,79,80,81,82,83].

Allergens		*Canis familiaris*								*Felis catus*								*Cavia porcellus*					P.s.	M.a.	M.m.	R.n.	*Oryctolagus cuniculus*				*Bos domesticus*												*Equus caballus*						E.a.	S.s.	C.h.	O.a.
		Can f 1	Can f 2	Can f 3	Can f 4	Can f 5	Can f 6	Can f 7	Can f 8	Fel d 1	Fel d 2	Fel d 3	Fel d 4	Fel d 5	Fel d 6	Fel d 7	Fel d 8	Cav p 1	Cav p 2	Cav p 3	Cav p 4	Cav p 6	Phod s 1	Mes a 1	Mus m 1	Rat n 1	Ory c 1	Ory c 2	Ory c 3	Ory c 4	Bos d 2	Bos d 3	Bos d 4	Bos d 5	Bos d 6	Bos d 7	Bos d 8	Bos d 9	Bos d 10	Bos d 11	Bos d 12	Bos d 13	Equ c 1	Equ c 2	Equ c 3	Equ c 4	Equ c 5	Equ c 6	Equ a 6	Sus s 1	Koza	Owca
** *Canis familiaris* **	Can f 1																																																			
	Can f 2																																																			
	Can f 3																																																			
	Can f 4																																																			
	Can f 5																																																			
	Can f 6																																																			
	Can f 7																																																			
	Can f 8																																																			
** *Felis catus* **	Fel d 1																																																			
	Fel d 2																																																			
	Fel d 3																																																			
	Fel d 4																																																			
	Fel d 5																																																			
	Fel d 6																																																			
	Fel d 7																																																			
	Fel d 8																																																			
** *Cavia porcellus* **	Cav p 1																																																			
	Cav p 2																																																			
	Cav p 3																																																			
	Cav p 4																																																			
	Cav p 6																																																			
**P.s.**	Phod s 1																																																			
**M.a.**	Mes a 1																																																			
**M.m.**	Mus m 1																																																			
**R.n.**	Rat n 1																																																			
**O.c.**	Ory c 1																																																			
	Ory c 2																																																			
	Ory c 3																																																			
	Ory c 4																																																			
** *Bos domesticus* **	Bos d 2																																																			
	Bos d 3																																																			
	Bos d 4																																																			
	Bos d 5																																																			
	Bos d 6																																																			
	Bos d 7																																																			
	Bos d 8																																																			
	Bos d 9																																																			
	Bos d 10																																																			
	Bos d 11																																																			
	Bos d 12																																																			
	Bos d 13																																																			
** *Equus caballus* **	Equ c 1																																																			
	Equ c 2																																																			
	Equ c 3																																																			
	Equ c 4																																																			
	Equ c 5																																																			
	Equ c 6																																																			
**E.a.**	Equ a 6																																																			
**S.s.**	Sus s 1																																																			
**C.h.**	Koza																																																			
**O.a.**	Owca																																																			
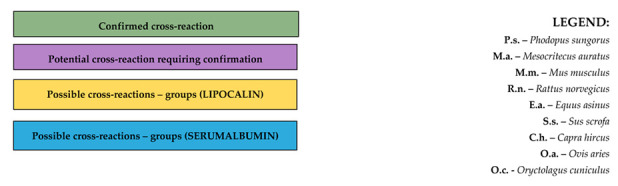																																																				

## 5. Are There Any Diagnostic Standards with Reference to Allergy to Furry Animals?

These days, the vast majority of data on the diagnostics and prevalence of sensitisation to furry animals focuses on the two most common pets—dogs and cats. It can be said with all certainty that the allergy to cats and dogs is a widespread phenomenon; therefore, it has a high impact on quality of life, and poses a challenge to healthcare professionals. Nevertheless, the increasing prevalence of uncommon pets means that diagnostic methods must “follow” the growing needs of the afflicted.

The guidelines from the European Academy of Allergy and Clinical Immunology (EAACI) released in 2022 address the issue of molecular diagnostics, and in the section concerning allergy to furry animals, devised by Marianne van Hage et al. [4], diagnostics in line with the “TOP-DOWN” approach are encouraged; the order of proposed research is presented in Figure 2.

The abovementioned guidelines also emphasise that in relation to standardised extracts in skin prick tests, it is possible to safely use cat allergenic extracts; however, dog allergenic extracts produced by various producers should be handled with caution as they demonstrate significant differences with reference to the main dog allergens. However, there are no data on the usefulness of horse allergen extract. According to the authors of the guidelines, total IgE concentration in persons allergic to furry animals has no great clinical relevance, and it is the determination of asIgE that allows for identifying the primary source of sensitisation. It was also noted that in complex conditions, it is possible to perform a nasal provocation test, in particular when referring to the diagnostics of dog allergies. In the case of a cat, such tests are not usually required to make a diagnosis and adapt the treatment plan, but they may be indicated in certain situations, such as polysensitisation or when discrepancies between the results of skin tests and IgE are observed.

With reference to the three most common pets in the human environment that might be responsible for causing clinical manifestation in allergy sufferers, the EAACI has drafted diagnostic schemes, presented below: Figure 3—for a cat; Figure 4—for a dog; and Figure 5—for a horse. The preparation of these schemes was based on the latest scientific reports that were adapted to the clinicians’ needs [84].

The treatment of pet sensitisation includes avoiding exposition to its allergens, symptomatic treatment, and, in some cases, immunotherapy and biological treatment, for example, with omalizumab. There are also several experimental treatment methods that are being taken into consideration and investigated. Most patients receive symptomatic treatment, dependent on the natural course of their disease [85]. Most patients with cat or dog allergy suffer from bronchial asthma and allergic rhinitis and should therefore receive appropriate treatment [86]. The detailed description of treatment opportunities remains outside of the scope of this review article, but definitely requires further analysis.

## 6. Summary

Allergy to furry animals, with its growing incidence, is of increasingly higher importance in the clinical practice of an allergist. In the course of diagnostics, we can use an arsenal of available allergy tests, both in vitro and in vivo, but it seems that it is molecular diagnostics that holds a special place, as it is the only one which can currently clearly address the questions concerning primary sensitisation and predicted cross-reactions and also enable taking accurate therapeutic measures. Nowadays, most publications refer to allergies to cats, dogs, and horses; yet, despite the availability of molecular tests, there is a lack of information on allergy to other furry animals, such as domestic cattle, rodents, or sheep. Further investigation of these groups will surely provide new facts and facilitate making clinical decisions when attending to patients sensitised to this group of allergens.

It is worth noting that the diagnosis of allergy to furry animals is largely similar to the diagnosis of other allergies to inhalant allergens, but there are several important differences that should be taken into account when comparing diagnostic schemes. Most diagnostic possibilities are limited by the significantly limited pool of animal allergens that are available in everyday medical practice, both in vitro and in vivo. Challenge tests are still only performed very rarely. Cross-reactivity between allergens originating from different species may be a challenge for clinicians. Newly described animal allergens as well as component-resolved diagnosis may help in predicting the natural course of the sensitisation. We hope that this review article will improve knowledge about the diagnosis of fur allergy, showing both possibilities and limitations, which will translate into an improvement in the quality of life of allergic patients.

## Figures and Tables

**Figure 1 jcm-13-03239-f001:**
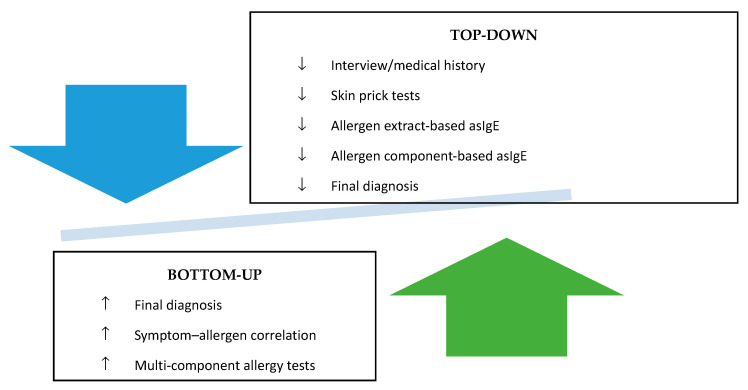
Example use of basic diagnostic models in allergology. asIgE—antigen-specific IgE.

**Figure 2 jcm-13-03239-f002:**
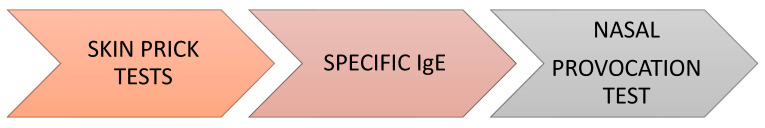
The order of diagnostic tests in allergy to furry animals—EAACI guidelines 2022 [84].

**Figure 3 jcm-13-03239-f003:**
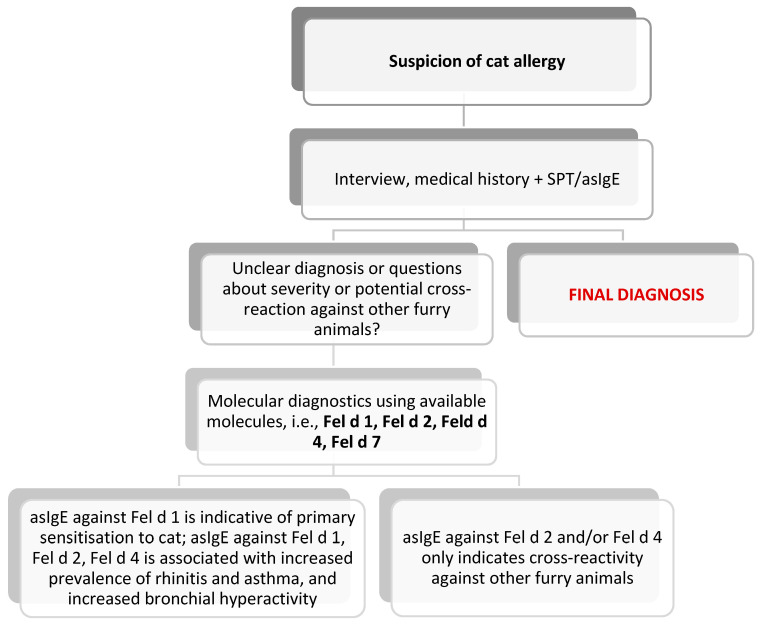
Diagnostic algorithm for cat allergy [84].

**Figure 4 jcm-13-03239-f004:**
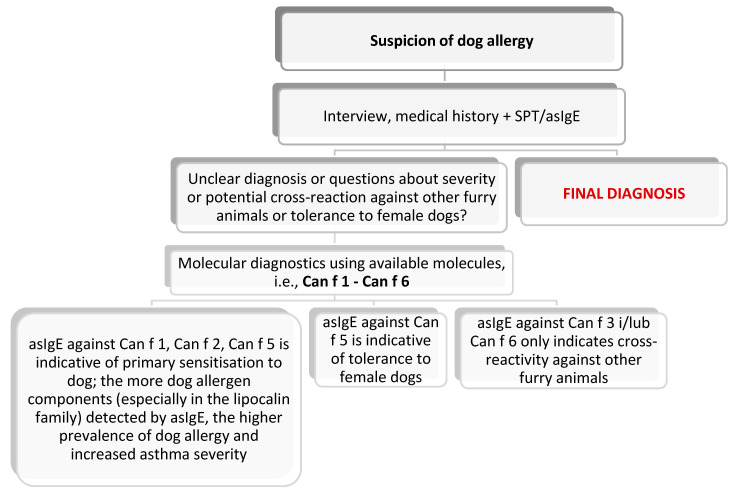
Diagnostic algorithm for dog allergy [84].

**Figure 5 jcm-13-03239-f005:**
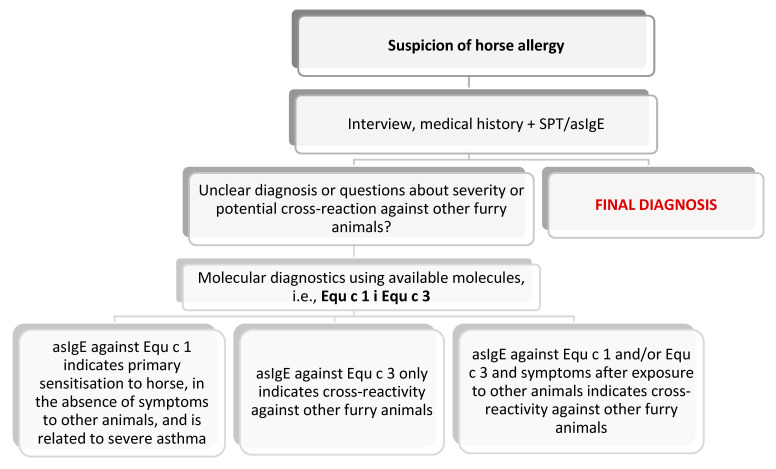
Diagnostic algorithm for horse allergy [84].

**Table 1 jcm-13-03239-t001:** Division of selected allergy tests into in vitro and in vivo [9,10].

Type of Tests		
In Vitro	In Vivo	Ex Vivo
Total IgE (tIgE) level	Atopy patch tests (APTs)	Basophil activation test (BAT)
Allergen-specific IgE (sIgE) level	Skin prick tests (SPTs)	Mast cell activation test (MAT)
Level of IgE specific for allergen molecules (molecular diagnostics)	Provocation tests	

**Table 2 jcm-13-03239-t002:** List of selected allergens for skin testing [14,15,16].

Allergen Extract	Allergopharma GmbH & Co. KG	Stallergenes S.A.	HAL Allergy B.V.
Animal-derived allergen extracts (selected)			
Hamster fur	X	X	
Dog fur	X	X	X
Rabbit fur	X	X	
Cat fur	X	X	X
Mouse fur	X		
Guinea-pig fur	X	X	
Horse fur	X	X	
Rat fur	X		
Cow fur	X		
Sheep wool	X	X	
Pig fur	X		
Goat fur	X	X	
Food allergen extracts (selected)			
Mutton/lamb	X	X	
Horsemeat	X		
Beef	X		
Pork	X	X	
Cow’s milk	X		
Goat’s milk		X	
Animal-derived mixture of allergen extracts			
Furs I	X		
Hamster			
Dog			
Rabbit			
Cat			
Guinea pig			
Furs II	X		
Horse			
Cow			
Sheep			
Pig			
Goat			
Mixture of food allergen extracts (selected)			
Meat I	X		
Mutton			
Beef			
Pork			

**Table 3 jcm-13-03239-t003:** Allergens of fur animals (airborne and food) currently available for diagnostics (based on the most popular in vitro diagnostic tests) [32,34,35,36,37,38,39,40,41,42,43,44,45,46,47].

		Singleplex				Multiplex					
Allergen Source (Allergen Number)	Extract/Molecule	ImmunoCAP (Thermo Scientific ^1^)	Immulite^2000^ (Siemens ^2^)	Noveos (HYCOR ^3^)	IDS (Immunodiagnostic Systems ^4^)	EUROLINE (Euroimmun ^5^)	POLYCHECK (Biocheck ^6^)	BioIC (AGNITIO ^7^)	RIDA qLine (R-Biopharm ^8^)	ALEX ^2^ (MADX ^9^)	ISAC*_E112i_* (Thermo Scientific ^1^)
Dog (*Canis familiaris*)											
Dog dander (E5)	EXTRACT	E	E	E	E				E		
Dog epithelium (E2)			E			E					
Dog dander (E5) and Dog epithelium (E2)	EXTRACT						E	E			
Dog urine (including Can f 5)	EXTRACT									E	
Can f Fel d 1-like	Secretoglobin									R	
Can f 1	Lipocalin	R	N	R	N					R	R
Can f 2	Lipocalin	R		R						R	R
Can f 3	Serum albumin	N	N	N						N	N
Can f 4	Lipocalin	R								R	R
Can f 5	Arginine esterase, prostatic kallikrein	R		R							R
Can f 6	Lipocalin	R								R	R
Cat (*Felis domesticus*)											
Cat dander (E1)	EXTRACT	E	E	E	E	E	E	E	E		
Fel d 1	Uteroglobin (chain 1)	R	N	R	N					R	R
Fel d 2 (E220)	Serum albumin	R	N	N						N	R
Fel d 4	Lipocalin	R		R						R	R
Fel d 7	Lipocalin	R								R	
Guinea pig (*Cavia procellus*)											
Guinea pig epithelium (E6)	EXTRACT	E	E	E		E	E		E		
Cav p 1	Lipocalin									R	
Cav p 2	Lipocalin										
Cav p 3	Lipocalin										
Cav p 4	Serum albumin										
Cav p 6	Lipocalin										
HAMSTERS (*Phodopus sungorus* (Phod s), *Mesocricetus auratus* (Mes a), *Cricetus cricetus* (Cir c)											
Hamster epithelium (E84)	EXTRACT	E (?)	E (Cir c)			E (?)	E (?)		E (Mes a)		
Phod s 1	Lipocalin									R	
Mes a 1	Lipocalin										
Mouse (*Muse musculus*)											
Mouse epithelium (E71)	EXTRACT	E	E	E		E	E				
Mouse urine proteins (E72)	EXTRACT	E	E	E							
Mouse serum proteins (E76)	EXTRACT	E	E	E							
Mus m 1	Lipocalin and urinary prealbumin									N	R
Rat (*Rattus norvegicus*, *Rattus rattus*)											
Rat epithelium (E73)	EXTRACT	E	E	E		E				E	
Rat urine protein (E74)	EXTRACT	E	E	E							
Rat serum protein (E75)	EXTRACT	E	E	E							
Rabbit (*Oryctolagus cuniculus*)											
Rabbit epithelium (E82)	EXTRACT	E	E	E		E	E		E		
Rabbit urine protein		E		E							
Rabbit serum protein		E		E							
Rabbit meat		E	E							E	
Ory c 1	Lipocalin									R	
Ory c 2	Lipocalin									R	
Ory c 3	Lipophilin/Uteroglobin									R	
Chinchilla (*Chinchilla chinchilla*)											
Chinchilla epithelium (E208)	EXTRACT	E									
Ferret (*Mustela furo*)											
Ferret epithelium (E217)	EXTRACT	E									
Gerbil (*Meriones unguiculatus*)											
Gerbil epithelium (E209)	EXTRACT	E	E								
Mink (*Neogale vison*, *Mustela lutreola*)											
Mink epithelium (E203)	EXTRACT	E	E								
Cow (*Bos domesticus*)											
Cow dander (E4)	EXTRACT	E	E	E	E	E					
Bos d 2	Lipocalin									R	
Cow’s milk	EXTRACT	E		E		E	E	E	E	E	
Cow’s milk whey	EXTRACT	E									
Cow’s whey	EXTRACT	E									
Boild milk	EXTRACT	E	E								
Milk powder						E					
Bos d 2	Lipocalin									R	
Bos d 4	α-lactalbumin	N	X	N	X	N	N	X	X	N	N
Bos d 5	β-lactoglobulin	N	X	N	X	N	N	X	X	N	N
Bos d 8	Casein	N	X	N	X	N	N		X	N	N
Beef	EXTRACT	E	E	E		E	E			E	
Bos d 6	Serum albumin	N		N		N	X		X	N	N
Bos d LTF	Laktoferyna	N				N					N
Horse (*Equus caballus*)											
Horse dander (E3)	EXTRACT	E	E	E	E	E			E		
Equine meat	EXTRACT									E	
Mare’s milk	EXTRACT	E								E	
Equ c 1	Lipocalin	R		R						R	R
Equ c 2	Lipocalin										
Equ c 3	Serum albumin									R	N
Equ c 4	Latherin									R	
Goat (*Capra hircus*)											
Goat epithelium (E80)	EXTRACT	E	E							E	
Goat’s milk	EXTRACT	E	E	E		E				E	
Sheep (*Ovis* spp., *Ovis aries*)											
Ship epithelium (E81)	EXTRACT	E	E				E			E	
Ship’s milk	EXTRACT	E		E						E	
Sheep’s whey	EXTRACT	E									
Mutton/Lamb (F88)	EXTRACT	E	E			E	E			E	
Pig (*Sus scrofa domesticus*)											
Pig epithelium (E83)	EXTRACT	E	E							E	
Pork	EXTRACT	E	E	E		E	E			E	
Sus s 1 (E222)	Serum albumin	N	X							R	
Camel (*Camelus bactrianus*, *domesticus*)											
Camel’s milk										E	
Mixes											
(E1, E3, E4, E5)	EXTRACTS	Mix	Mix	Mix							
(E1, E5, E6, E87, E88)	EXTRACTS	Mix	Mix								
(E6, E82, E84, E87, E88)	EXTRACTS	Mix	Mix								
Rat epithelium, serum and urine protein (E87)	EXTRACTS	E	Mix	Mix							
Muse epithelium, serum and urine protein (E88)	EXTRACTS	E	Mix	Mix							
(E4, E80, E81)						Mix					
rCan f 1 nCan f 3 rCan f 5	MOLECULES			Mix							

Legend: Symbols: E—extract; E (?)—the species was not specified in detail by the manufacturer; R—recombinant molecule; N—native molecule; X—the type of molecule was not specified by the manufacturer; Mix—mix. Manufacturers: ^1^ Thermo Scientfic, Waltham, MA, USA; ^2^ Siemens Healthcare Diagnostics, Forchheim, Germany; ^3^ HYCOR Biomedical, LLC., Garden Grove, CA, USA; ^4^ Immunodiagnostic Systems Ltd., Boldon Colliery, UK; ^5^ Euroimmun, Lubeca, Germany; ^6^ Biocheck GmbH, Münster, Germany; ^7^ Agnitio Science Technology, Hsinchu, Taiwan; ^8^ R-Biopharm AG, Darmstadt, Germany; ^9^ MacroArray Diagnostics GmbH, Vienna, Austria.

**Table 4 jcm-13-03239-t004:** Commercially available allergens for use in basophil activation tests [54].

Allergens—The BAT	
BAG-E1	Cat epithelium
BAG2-FELD1	rFEL D 1, cat
BAG-E2	Dog epithelium
BAG-F1	Egg white
BAG-F75	Egg yolk
BAG-F2	Cow’s milk
BAG-F76	Alpha-lactalbumin
BAG-F77	Beta-lactoglobulin
BAG-F78	Casein
BAG-F27	Beef

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
