# Peer review of "Diagnostics of Allergy to Furry Animals—Possibilities in 2024"

_jcm, 2024, doi:10.3390/jcm13113239_

Round 1

Reviewer 1 Report

Comments and Suggestions for Authors

I have reviewed the article entitle “Diagnostics of Allergy to Furry Animals—What are the Opportunities in 2024 " with a great interest.

It focuses on an important topic which there are still gaps both in the literature and current clinical practice.

However, although it is not a systematic review, I think providing a structured search query and database information could improve the quality of the paper. Hence, it could provide a broader approach to the current literature and helps authors to be make sure that all related important references have been included to the manuscript.

Another major issue is the incoherence between main scope and context of the paper. Although the main scope of the paper is furry animal allergy, there are some chapters which were focused on allergy diagnosis in general practice, with a little focus on furry animal allergy (please see Section 3).

Lastly, related references should be cited where the information was taken throughout the manuscript.

I have some other points below, hope to improve the quality of authors' work and to increase its dissemination.

Abstract

Comment 1: In abstract, authors emphasize the lack of data about “uncommon pets”, however, there is very little information about uncommon pets in the main manuscript.

Section 2.

Comment 2: Could the authors add related references with regard to  “Top-down” and “Bottom-up” strategies? For example, it has been clearly explained in Molecular Allergology User Guide 2.0. (Dramburg S et al.,  EAACI Molecular Allergology User's Guide 2.0. Pediatr Allergy Immunol. 2023 Mar;34 Suppl 28:e13854. doi: 10.1111/pai.13854. PMID: 37186333.)

Comment 3: Does authors think including “U-shape approach” which has been also suggested in MAUG 2.0? (doi: 10.1111/pai.13854)

Section 3.

Comment 4: In this section, authors focus on the skin prick test methodology detailly in allergy practice. It is very important; however, it is not the main aim of the current paper. Since the main scope of the review is on furry animal allergies, do authors think to shorten this part? Instead, I suggest authors could focus on advantages and disadvantages of skin prick test with a focus on diagnosis of furry animal allergies? For example, unstandardized dog dander extracts could be mentioned here (Curin et al., 2011, doi: 10.1159/000321113)

Comment 5: Information in paragraph starting with “The prick-by-prick tests” does not supported by related references.

Comment 6: Sentence starting as “ From my personal experience..” might not be compatible with academic writing. I suggest revising this part according to suitable academic writing customs (maybe supporting with a previous case report/publication).

Comment 7: “These tests are characterised by  good sensitivity and specificity, and the negative predictive value.” For me, it is not clear the referred test (for which allergen?). The related reference should be added as well.

Section 4.

Comment 8: Definitions and general information about cross-reactivity have not been supported by suitable references.

Comment 9: For cross-reactivity, high sequence identity is also important in addition to structural similarity . Could authors elaborate on that in paragraph starting from ”It is argued that two..”?  Authors suggest that “Current research investigate rather the spatial structure of the allergen, its binding sites with various antibodies and the resulting conformational simi larity.”. Please add the related reference where the information was taken.

Comment 10: “Several studies...”-Please include the mentioned studies in the reference list. Authors could also revisit some important studies conducted by Van Hage and Kondradsen group (e.g., a recent review by same group doi: 10.1016/j.molimm.2023.03.003)

Comment 11: Hemmer et al. (10.1111/all.14885)doi: 10.1111/all.14885.) has investigated the cross-reactivity patterns and their clinical relevance. Since it is in scope of this section, main findings could be mentioned here as well.

Section 5

Comment 12: In this section, authors address MAUG’s approach for allergy to furry animals. I believe this is very important, however, it looks repetitive-I could not find any additional information. This section could be enriched with other papers/or with a discussion based on the given information based on MAUG.

Tables and Figures

Comment 13: Table 2 focus on SPT drug discontinuation time before SPT. While this is an important information, this table does not coherent with the main scope of the review.

Minor comments:

Comment 14: To the best of my knowledge, asIgE is not a very commonly used abbreviation in the nomenclature.

Comments on the Quality of English Language

-

Author Response

Dear Reviewer 1,

Thank you for the time and effort it took to evaluate our manuscript. We feel that due to your insight the manuscript is now much more clear and interesting. I hope that the improvements we have made will met up with your expectations and you will find the review article worth publishing.

As you pointed out this article does not met the criteria for systematic review, but the query and database information was included in the appropriate section of the manuscript. The sections that are to long and outside the scope of the subject of furry animals were deleted or modifies. The references were modified, updated and corrected.

In the abstract the “uncommon pets” was found not to be appropriate, was changed to, in fact, other pets than cats and dogs.

“Top-down” and “Bottom-up” strategies were correctly referenced. MAUG’s approach for allergy was presented in detail.

“U-shape approach” was included.

The skin prick test methodology in now more focused.

The prick-by-prick tests” in now supported by correct reference.

 The language was modified and the personal experiences mentioned were appropriately referenced, as well as positive and negative predictive value information and cross reactivity.

Interesting findings by Hemmer et al. (10.1111/all.14885)doi: 10.1111/all.14885.) were included in the manuscript.

asIgE abbreviation was clearly explained.

Both linear and conformational epitopes are explained In the context of cross reactivity.

Interesting refrence from Van Hage and Kondradsen group was included in the review article.

Table 2 was not coherent with the main scope of the review and was removed.

I hope that the revised version of the article will met up with your expectations. We feel that thanks to your insight the manuscript is now much more interesting and it improved a lot. Thank you for your help in making our work better. Hopefully you will find revised version suitable for publication. If however you feel that there are other issues to address please do not hesitate to contact us.

Kind regards

Tomasz Rosada, on behalf of the authors

Reviewer 2 Report

Comments and Suggestions for Authors

Dear authors, this review about diagnostics of allergy to furry animals is interesting and puts the lights on alergies which are not common and with some diagnostic challenges.

There are only some minor flaws to assess:

- writing (commented in the proper section);

- The references could be improved including other international journals and also the reference style could be improved;

- Some tabs are not centered;

- There's a font error in line 156.

Comments on the Quality of English Language

- The writing could be improved maybe through the help from a native speaker;

Author Response

Dear Reviewer 2,

Thank you for the time it took to evaluate our work. We are pleased to hear you found the manuscript interesting. As requested the manuscript underwent editing by an English speaker. References were updated and the reference style was improved. The tables were edited, for clarity.

I hope that in the current form you will find our manuscript worth publishing.

Kind regards

Tomasz Rosada, on behalf of the authors

Reviewer 3 Report

Comments and Suggestions for Authors

The study has a good  harmony and contains  valuable  findings. This study is the most comprehensive review in regarding field. 

Since the allergy diagnostic tests for most allergens are almost the same, the only thing that seems to be what is explsin is the difference between the diagnosis of furry animal allergens and other allergens?

Author Response

Dear Reviewer 3,

Thank you for your review and kind words. It is very important for us, that you found our work comprehensive and valuable. Hopefully now, after the review, you will find the manuscript worth publishing.

Kind regards

Tomasz Rosada, on behalf of the authors

Reviewer 4 Report

Comments and Suggestions for Authors

Allergy to furry animals takes an increasingly higher position in the clinical practice of an allergist.

Authors, in this review well outlined, by ranging allergy tests from skin tests, through specific IgE, to molecular panels and challenge tests. 

There many self citation, that should be evaluated by the authors

Author Response

Dear Reviewer 4,

Thank you for your review. We are pleased that you found our work well outlined. The self-citations were removed and the references were updated.

I hope that in the current form you will find the manuscript worth publishing. Once again thank you for your help in making this manuscript better.

Kind regards

Tomasz Rosada, on behalf of the authors

Reviewer 5 Report

Comments and Suggestions for Authors

The work is presented concisely and minor corrections are necessary:

1. State with H2 blockers that it is necessary to include PPIs in patients with CKD because they worsen kidney disease!

2. In Table 4, change the red color to something less striking and more pleasing to the eye!

3. In Table 6, pink is barely noticeable, so it will be replaced by green or purple

Author Response

Dear Reviewer 5,

Thank you for the time and effort it took to evaluate our work. The Table 4 and Table 6 were modified as requested and now are much more visible and readable. Thank you for this suggestion.

As for H2 blockers, PPIs and CKD –the main scope of the manuscript is the diagnostic procedures in furry animals sensitization. We tried to focus the attention on this subject, other aspects of sensitization were limited. The information provided and the focus on H2 blockers, PPIs and CKD is very much of interest. In follow up article, that we are preparing, regarding the treatment of furry animals allergy, this topic might be of great interest. Thank you for pointing it out.

We hope that now, after review, in the current form, you will find the manuscript interesting and worth publishing.

Kind regards

Tomasz Rosada, on behalf of the authors
